# An Update on the Current and Emerging Use of Thiazolidinediones for Type 2 Diabetes

**DOI:** 10.3390/medicina58101475

**Published:** 2022-10-17

**Authors:** Rosaria Vincenza Giglio, Nikolaos Papanas, Ali Abbas Rizvi, Marcello Ciaccio, Angelo Maria Patti, Ioannis Ilias, Anca Pantea Stoian, Amirhossein Sahebkar, Andrej Janez, Manfredi Rizzo

**Affiliations:** 1Department of Biomedicine, Neuroscience, and Advanced Diagnostics, University of Palermo, 90127 Palermo, Italy; 2Department of Laboratory Medicine, University Hospital “Paolo Giaccone”, 90127 Palermo, Italy; 3Diabetes Centre, Second Department of Internal Medicine, Democritus University of Thrace, University Hospital of Alexandroupolis, 68132 Alexandroupoli, Greece; 4Department of Medicine, University of Central Florida College of Medicine, Orlando, FL 32827, USA; 5Promise Department, School of Medicine, University of Palermo, 90133 Palermo, Italy; 6Department of Endocrinology, Diabetes and Metabolism, Elena Venizelou Hospital, 11521 Athens, Greece; 7Faculty of Medicine, Diabetes, Nutrition and Metabolic Diseases, Carol Davila University, 050474 Bucharest, Romania; 8Department of Biotechnology, School of Pharmacy, Mashhad University of Medical Sciences, Mashhad 1696700, Iran; 9Department of Endocrinology, Diabetes and Metabolic Diseases, University Medical Centre, University of Ljubljana, 1000 Ljubljana, Slovenia

**Keywords:** type 2 diabetes mellitus, pioglitazone, cardiovascular risk, metabolic syndrome

## Abstract

Guidelines have increasingly stressed the concept that adequate glycemic control is required to prevent or decrease the macro- and microvascular complications of type 2 diabetes mellitus (T2DM). PPAR-gamma agonists (“glitazones”) are no longer prioritized due to their effects on heart failure. However, the association between these drugs and innovative therapies could be a valuable tool to attenuate the risk factors of the metabolic syndrome. Glitazones are used for the treatment of diabetes and associated comorbidities. There is substantial scientific evidence demonstrating the effect of glitazones at a cardiometabolic level, as well as on hematological and neurological pathologies that point to their usefulness. The use of glitazones has always been controversial both for the type of patients who must take these drugs and for the side effects associated with them. Unfortunately, the recent guidelines do not include them among the preferred drugs for the treatment of hyperglycemia and rosiglitazone is out of the market in many countries due to an adverse cardiovascular risk profile. Even though real-life studies have proven otherwise, and their pleiotropic effects have been highlighted, they have been unable to achieve primacy in the choice of antihyperglycemic drugs. It would be appropriate to demonstrate the usefulness of pioglitazone and its therapeutic benefit with further cardiovascular safety studies.

## 1. Introduction

Important intervention studies to prevent or reduce the impact of micro/macrovascular consequences in Type 2 Diabetes Mellitus (T2DM) have established the importance of adequate glycemic control [1,2,3,4,5]. Professional societies have promulgated guidelines arguing that the therapeutic goal, pursued with insulin, sulfonylureas, metformin and acarbose, in addition to thiazolidinediones (TZD) or glitazones, in monotherapy or in combination with metformin should be implemented with innovative therapies that have both pleiotropic and glucose-lowering effects [6]. Unfortunately, in the latest guidelines the TZDs have not been emphasized due to their dubious effects on heart failure. However, use of these drugs in concert with innovative therapies could likely be useful in mitigating the glucometabolic complications of T2DM.

## 2. Current Approaches in the Management of T2DM

T2DM is the result of two underlying pathophysiological abnormalities that disrupt homeostasis, namely insulin resistance (IR) and dysfunction of beta cells [7]. However, there are several factors in insulin deficiency and its functional incapacity and the contribution of these components in glycemic balance is of crucial importance [8]. In T2DM, IR is strongly determined by α cells (hyperglucagonemia), adipocytes (fat lipolysis), kidneys (high glycemic reabsorption), the gastro-intestinal system (resistance and/or deficiency of incretin), the brain (IR), and by complicated associations of these metabolic dysfunctions with genes linked with T2DM [9]. In addition, IR is closely related to non-alcoholic fatty liver disease (NAFLD); indeed, increased hepatic lipid storage is an early abnormality in IR patients and this has a particular importance in daily clinical practice since NAFLD is a chronic liver disease that affects up to one-third of the adult population in industrialized countries [10].

Lifestyle changes of diabetic individuals are important in combination with drug interactions to ameliorate the general state of patient’s health. Sport activity is significantly related with improved blood glucose concentrations among T2DM patients. Diabetes complications have been shown to be effectively delayed if moderate, daily physical activity is performed; decreased sedentary behavior is therefore fundamental in T2DM, and exercise is helpful. [11]. Furthermore, daily exercise is also recognized to ameliorate glycated hemoglobin (HbA_1c_) and insulin sensitivity [12].

Stress management strategies represent an innovative approach to effectively prevent the incidence of T2DM. Constant exposure to stressors is implicated in the increased risk of developing T2DM [13]. Long-term stress is connected with dysregulated glycemic metabolism and neuro-endocrine abnormalities through chronic, low-grade inflammation. Most of the risk factors involved in diabetes are impacted by psychological conditions, which influence the release of glucose and the lipid component in blood vessels, the release of inflammatory cytokines, and hypertension [14]. Therefore, the main factor causing the increased T2DM risk is certainly the allostatic load of the body following long-term exposure to psychological insults [15]. Increased stress levels are linked to low therapy compliance and glycemic balance in subjects with T2DM [16].

The American Diabetes Association guidelines, together with International recommendations from other prestigious societies and expert panels, suggest that for patients with T2DM and CardioVascular Disease (CVD) or Chronic Kidney Disease (CKD) the addition of Sodium–Glucose Cotransporter 2 (SGTL2) inhibitors and/or Glucagon-Like Peptide 1 Receptor agonists (GLP1-Ras) should be considered as first-line therapy [17,18,19]; indeed, these agents should be used not only to lower HbA_1c_, but also for improvement of the patient’s general cardio-renal-metabolic condition [19]. SGTL2 inhibitors are ideal candidates to be combined with other therapies, to optimize the hypoglycemic effect while also improving other cardiometabolic risk parameters such as blood pressure and plasma lipids [17,18,19]. The use of these molecules in association with GLP1-RAs has also been recommended for patients at high cardiovascular risk in association with conventional therapies precisely for the improvement of various cardiovascular risk factors, although the adoption of these novel anti-diabetic treatments remains suboptimal in most countries [20]. GLP-1 agonists exert a very important pleiotropic effect in improving cardiovascular health [21,22] and some of them, such as liraglutide, are able to improve subclinical atherosclerosis by reducing atherogenic small dense low-density lipoproteins (LDL) [23]. This important molecular effect has been proposed as a key mechanism by which GLP-1 agonists are able to have a positive cardiovascular outcome [24]; indeed, the oxidation in the intima of small dense LDL is considered the first intra-vessel alteration of the complex atherosclerotic cascade, leading to the formation of foam cells and, ultimately, of atherosclerotic plaque [25]. In addition, the predominance of small dense LDL is strictly linked to endothelial dysfunction, predicting future cardiovascular events [26,27].

## 3. Current and Emerging Role of Thiazolidinediones

Peroxisome Proliferator-Activated Receptors (PPARs) represent nuclear receptors expressed throughout most of the body, including adipose tissue, liver, skeletal muscle, heart and kidney, and are deputies to the regulation of transcription of genes involved in gluconeogenesis, lipid transport and fatty acid oxidation [28]. The three PPARs (α, β/δ and γ) have a different tissue distribution: heart, skeletal muscle, brown fat, kidney, colon, liver randomly. Some PPAR agonists are prescribed for the management of T2DM, such as rosiglitazone and pioglitazone [28]. Table 1 and Table 2 summarize landmark clinical studies and major results achieved with thiazolidinediones. Figure 1 illustrates the mechanisms and pleiotropic effects of thiazolidinediones.

Pioglitazone is a PPARγ and PPARα agonist whichmodulatesthat modulates important genes involved in lipid metabolism. It acts on pathways that result in an increase in LPL synthesis with reduction of triglyceride content in lipoproteins; increase in the synthesis of fatty acid transporters on the membrane; increase in beta and omega oxidation of fatty acids; ketogenesis; enhancement of amino acid catabolism, urea cycle and gluconeogenesis; and increase in HDL-C synthesis, especially owing to increased expression of Apo AI and AII [29]. In the Pioglitazone vs. Vitamin E vs. Placebo for Treatment of Non-Diabetic Patients With Nonalcoholic Steatohepatitis (PIVENS) trial (ClinicalTrials.gov Identifier: NCT00063622), pioglitazone treatment resulted in a positive decrease of in liver enzymes (aspartate aminotransferase -AST, alanine aminotransferase -ALT), fatty liver and lobular inflammation in NonAlcoholic SteatoHepatitis (NASH) patients, possibly due to an improvement in small dense lipoproteins [30].

In patients suffering from glycemic disorders and NASH, pioglitazone ameliorates steatosis, inflammation, and liver fibrosis [31]. The favorable actions of pioglitazone on lipoproteins (an increment in large Low-Density Lipoproteins (LDL) along with a little but significant increment in LDL-C) [32] could be connected to its capacity to decrease and reverse IR. The modest increment in LDL-C does not increase CVD, partly owing to the beneficial effects on triglyceride (TG) and high-density lipoprotein cholesterol (HDL-C).

Pioglitazone reduces and stabilizes the atherosclerotic plaque. It improves plaque nucleus and reduces plaque inflammation [33]. Pioglitazone and rosiglitazone have a divergent cardiovascular safety profile, with the latter not available in most countries worldwide for a reported increase in cardiovascular events. It has been postulated that the increase in atherogenic lipoproteins in fasting and post-prandial status with rosiglitazone, and not with pioglitazone, may help to explain the difference in their cardiovascular actions [32,34]. Pioglitazone and rosiglitazone produce different effects on lipids: pioglitazone has a positive effect on triglycerides, HDL cholesterol, non-HDL cholesterol and on the size and concentration of LDL particles; rosiglitazone increases LDL-C levels, compared to pioglitazone (but does not change the LDL: HDL cholesterol ratio) and does not have a significant action on triglycerides [35]. The importance of novel therapeutical approaches for raising HDL-C and augmenting HDL particle functionality has been emphasized [36], in order to decrease the number of patients with low HDL concentrations, since this is strictly associated with cardiovascular risk [37]. Further, when a safety issue exists, there is always the option to use natural products with proven benefit, such as the case of red yeast rice supplementation for lipid alterations [38].

Both pioglitazone and rosiglitazone cause favorable modifications of the chemical and physical characteristics of LDL which are made less dense and, therefore, potentially less atherogenic. Indeed, small dense LDL have emerged as an independent cardiovascular risk factor in many high-risk populations including those with T2DM [39,40], promoting atherosclerotic plaque progression [41]. In DM subjects with coronary artery disease, treatment with pioglitazone resulted in a significantly lower rate of progression of coronary atherosclerosis [42] and slowed progression of CIMT compared with glimepiride [43].

Unfortunately, pioglitazone has shown a negative impact on heart failure. Indeed, an increase in the hospitalization rate in subjects with Familial Hypercholesterolemia (HF) associated with the use of pioglitazone has been observed, which appeared to be limited to 30 patients who had a history of overt CVD [44]. The increase in edema in patients taking pioglitazone therapy due to increased vascular permeability (effects on the sodium channel of the renal tubular epithelium and other effects in the collecting ducts), determines the increase in heart failure (HF) rates [44]. Edema could be the consequence of increased insulin-induced vasodilation, direct vasoactive effect, endothelin inhibition, and increased VEG production [45]. Lower doses of pioglitazone may be recommended to avoid fluid retention. Low-dose pioglitazone therapy may show the same degree of improvements in glucose and lipid metabolism, fatty liver, insulin resistance, and adiponectin as standard and high-dose pioglitazone therapy, but may also show negative effects on weight gain, edema and heart failure compared with standard and high-dose pioglitazone therapy [46]. Pioglitazone should be prescribed with caution in subjects with symptomatic HF, given that fluid overload is particularly deleterious in such subjects. The protective effect on the kidneys is robust, and varies depending on the different categories of baseline renal function. The decrease in albuminuria in studies with pioglitazone suggests that in addition to the decrease in glucose, insulin and blood pressure [47], pioglitazone may have renoprotective properties by acting directly on PPARγ receptors in the kidney [48]. In the latter, pioglitazone may decrease expression of deleterious growth factors, increase nitric oxide, reduce endothelin-1 [49], improve small vessel and endothelial function, and reduce Renin–Angiotensin System (RAS) activation [50]. Large reductions in kidney disease progression are observed regardless of baseline renal profile.

The PROspective pioglitAzone Clinical Trial In macroVascular Events (PROACTIVE) (*ClinicalTrials.gov Identifier: NCT00174993*) evaluated the effects of pioglitazone on morbidity and macrovascular mortality in T2DM patients with macrovascular disease. The latter was defined as: myocardial infarction or stroke before 6 months prior to admission; percutaneous coronary intervention (PCI) or coronary artery bypass graft (CABG) 6 months prior to enrollment; acute coronary syndrome (ACS) 3 months prior to enrollment; clinical evidence of coronary artery disease (CAD) or peripheral arterial disease (PAD). A significant decrease was observed with pioglitazone in the main secondary composite endpoint of all-cause mortality, non-fatal myocardial infarction (MI) and stroke (HR: 0.84, 95% CI: 0.72–0.98; *p* = 0.027). Heart failure deaths were not increased with pioglitazone (*p* = 0.634) [51]. This agent should be borne in mind in T2DM subjects with an established history of CVD.

A long-term therapeutic study, Thiazolidinediones Or Sulphonylureas and Cardiovascular Accidents Intervention Trial (TOSCA.IT) *(ClinicalTrials.gov Identifier: NCT00700856)* [52] showed that the prevalence of CVD was analogous with sulfonylureas (principally gliclazide and glimepiride) and pioglitazone, both added to metformin (HR: 0.96, 95% CI 0.74–1.26, *p* = 0.79). Importantly, lower hypoglycemia rates were seen with pioglitazone (*p* < 0.0001) [52].

The PPARγ agonist rosiglitazone increases serum lipids (LDL-C, non-HDL-C and TG levels) [35]. The Rosiglitazone Evaluated for Cardiac Outcomes and Regulation of glycemia in Diabetes (RECORD) *(ClinicalTrials.gov Identifier: NCT00379769)* study investigated 4447 T2DM subjects randomized to add-on rosiglitazone or to a combination of common hypoglycemic drugs (metformin and sulphonylurea) [53]. Cardiovascular death or cardiovascular hospitalization did not differ between the groups (HR: 0.99, 95% CI: 0.85–1.16, *p* = 0.93). No differences were seen in cardiovascular death (HR: 0.84, 95% CI: 0.59–1.18), myocardial infarction (HR: 1.14, 95% CI: 0.80–1.63) and stroke (HR: 0.72, 95% CI: 0.49–1.06) [49]. HF leading to hospitalization or death was more frequent with rosiglitazone (HR: 2.10, 95% CI: 1.35–3.27, *p* = 0.0010). Add-on rosiglitazone in T2DM increases the risk of HF and certain bone fractures, without improving cardiovascular morbidity or mortality [54]. Rosiglitazone should be prescribed with caution in subjects with or at risk of developing CVD.

Prediabetic status impairs prognosis in acute coronary syndrome. Obesity increases visceral fat, impairing metabolism. The precise pathogenesis of weight increment linked with TZD therapy is undetermined. In humans, variations that produce constitutively effective PPARγ (the genetic alternative of TZD application) result in grave obesity while dominant negative PPARγ variation causes partial lipodystrophy [55]. PPARγ is abundantly expressed in T2DM, these agents reduce HbA_1c_ by 1.5% and reduce HbA_1c_ to less than 7% in 30% of patients due to reduced muscle IR [56]. TZDs improve insulin sensitivity by increasing adiponectin [56]. This in turn promotes fatty acid oxidation, decreases glucose production and reduces hepatic fatty acids and triglycerides [57].

TZDs increase subcutaneous fat and reduce visceral fat [54]. Despite weight gain, they reduce liver fat [58] with encouraging results for NASH therapy [59]. Average weight gain is generally dose-dependent, especially when these drugs are combined with insulin, and is caused by increased subcutaneous fat and/or water retention and related to glycemic improvement [60]. The Fatty Liver Improvement with Rosiglitazone Therapy (FLIRT) (*ClinicalTrials.gov Identifier: NCT00492700*) study [61] showed favorable effects of this agent on insulin sensitivity and ALT normalization and a significant 30% reduction in hepatic steatosis vs. placebo [61].

The PIVENS study evaluated the benefit of pioglitazone and vitamin E therapy in the management of Non-Alcoholic Fatty Liver Disease (NAFLD) in subjects without T2DM. The groups treated with vitamin E and pioglitazone showed a significative reduction in fatty liver and lobular inflammation, and resolution of NASH (47% vs. 21%, *p* < 0.001), without significant amelioration in the fibrosis score [62]. There is a fourth class of double PPAR agonists, the glitazar (aleglitazar, muraglitazar and tesaglitazar), which bind to both α and γ PPAR isoforms and are beneficial for the treatment of metabolic syndrome. Saroglitazar, approved for diabetic dyslipidemia, is a potential therapeutic option for NAFLD. It leads to significant improvement in transaminases, hepatic stiffness and hepatic fibrosis in NAFLD patients with diabetic dyslipidemia. These changes were associated with a favorable impact on glucose, lipid and anthropometric parameters [63].

Women with previous Gestational Diabetes Mellitus (GDM) [64] carry a higher risk of future diabetes, especially 5–10 years after delivery [65]. In non-diabetic women with insulin resistance, β-cell decline was prevented by 3-year pioglitazone therapy [66]. The use of pioglitazone has beneficial effects in delaying the decline of insulin secretion, thus reducing the likelihood of requiring additional drugs to maintain adequate glycometabolic control. Glitazones have been shown to be able to maintain a good metabolic control with durability over time due to a protective effect of the drug at the beta-cell level in the type 2 diabetic patient [67].

Women who had taken troglitazone in the TRIPOD trial exhibited significant changes in β-cell function [68]. Insulin resistance was improved, but only transiently during thiazolidinedione treatment [68]. The initial insulin reduction affected diabetes risk: diabetes incidence was lowest among those with the greatest therapy-induced reduction [68]. Insulin sensitivity was significantly increased solely among those not progressing to T2DM [68].

**Table 1 medicina-58-01475-t001:** Summary of landmark clinical studies involving thiazolidinediones.

Trial Name and Acronym	Short Description of Methods	Subjects and Duration	Results
Pioglitazone vs. Vitamin E vs. Placebo for Treatment of Non-Diabetic Patients With Nonalcoholic Steatohepatitis (PIVENS) *ref. [30]*	Determine if therapy with pioglitazone or vitamin E will lead to an improvement in liver histology in non-diabetic adult patients with NASH	A total of 247 participants with improvement in NAFLD activity defined by change in standardized scoring of liver biopsies at baseline and after 96 weeks of treatment	Significant reduction in ALT, AST, fatty liver and lobular inflammation in patients with NASH, possibly due to an improvement in small dense lipoproteins
PROspective pioglitAzone Clinical Trial In macroVascular Events (PROACTIVE) *ref. [51]*	Determine whether pioglitazone, once daily, can delay the time to death, heart attack, acute coronary syndrome, heart bypass surgery, stroke, leg bypass surgery or amputation in patients with type 2 diabetes	A total of 4373 participants treated with pioglitazone in combination with other medications for glycemic management; pioglitazone might reduce the incidence of macrovascular events associated with this disease compared with placebo; 4 years	Significant reduction was observed with pioglitazone in the main secondary composite endpoint of all-cause mortality, non-fatal MI and stroke; the number of deaths from heart failure was similar in both the pioglitazone and the placebo groups
Thiazolidinediones Or Sulphonylureas and Cardiovascular Accidents Intervention Trial (TOSCA.IT) *ref. [52]*	Evaluate the effects of add-on pioglitazone as compared with add-on a SU on the incidence of cardiovascular events in T2DM patients inadequately controlled with metformin; compare the two treatments in terms of glycemic control, safety, and economic costs	A total of 3371 T2DM patients; primary composite endpoint of all-cause mortality, non-fatal MI (including silent MI), non-fatal stroke, and unplanned coronary revascularization; secondary outcomes. Principal secondary outcome: a composite ischemic endpoint of sudden death, fatal and non-fatal acute MI (including silent MI), fatal and non-fatal stroke, major amputations (above ankle), endovascular or surgical intervention on the coronary, leg or carotid arteries	The incidence of cardiovascular events was similar with sulfonylureas and pioglitazone as add-on treatments to metformin but with fewer hypoglycemic events in the latter
Rosiglitazone Evaluated for Cardiac Outcomes and Regulation of glycemia in Diabetes (RECORD) study *ref. [53]*	The study is being performed to monitor the incidence of cancer and bone fractures in RECORD patients for a period of 4 years after the end of the main RECORD study (2008–2012)	In this study, patients inadequately controlled on background metformin will be randomized to receive, in addition to metformin, either rosiglitazone or a SU in a ratio of 1:1; patients inadequately controlled on background SU will be randomized to receive, in addition to SU, either rosiglitazone or metformin in a ratio of 1:1; equal numbers of patients receiving background metformin and SU at entry will be entered into the study	The primary outcome of cardiovascular death or cardiovascular hospitalization was similar in both groups and there was no significant difference for cardiovascular death, MI and stroke; there was a significant increase in the rate of heart failure that resulted in hospitalization or death with rosiglitazone compared to the active controls
Fatty Liver Improvement with Rosiglitazone Therapy (FLIRT) study *ref. [61]*	Find out whether treatment with rosiglitazone improves the state of the liver and related blood markers in patients with NASH	A total of 64 patients with biopsy proven NASH will be randomized to receive either rosiglitazone 8 mg/day or placebo for one year; after one year of treatment patients will undergo a liver biopsy then a 4-month follow off treatment	Improvement in insulin sensitivity and a normalization of ALT levels that were four times greater than the placebo group; a significant 30% decrease in hepatic steatosis was observed; there was no significant histological improvement in the rosiglitazone group compared to the placebo group
A Randomized Trial to Slow the Progression of Diabetes (TRIPOD) *ref. [68]*	Test whether an evidence-based, low-cost mobile DMP, with or without an incentive program grounded in economic theory (M-POWER Rewards), can effectively and cost-effectively improve health outcomes for adults with type 2 diabetes	A total of 262 participants; a 52-week, three-arm randomized controlled trial to evaluate whether an evidence-based, low-cost mobile DMP, with or without an incentive program grounded in economic theory, can effectively and cost-effectively improve outcomes for adults with diabetes	Insulin resistance was improved during the thiazolidinedione treatment periods

ALT: ALanine aminotransferase; AST: ASpartate aminotransferase; DMP: Diabetes Management Package; MI: Myocardial Infarction; NAFLD: Non-Alcoholic Fatty Liver Disease; NASH: Non-Alcoholic SteatoHepatitis; SU: Sulfonylurea; T2DM: Type 2 Diabetes Mellitus.

**Table 2 medicina-58-01475-t002:** Use of the thiazolidinediones in type 2 diabetes and cardiometabolic disorders.

Monotherapy	Combination Treatment with Other Antidiabetic Agents	Use in T2DM and Atherosclerotic Disease
Improves steatosis, inflammation, liver fibrosis Reduce and reverse insulin resistance May decrease renal expression of growth factors implicated in the pathogenesis of diabetic nephropathy, increase renal bioavailability of nitric oxide, decrease renal expression of endothelin-1, improve renal microcirculation and endothelium function and downregulate the activity of the RAS Decrease muscle insulin resistance and increase levels of adiponectin TZDs are able to reduce intrahepatic fat accumulation with promising results regarding its efficacy in the treatment of patients with NASH Pioglitazone could also restore menstrual cycles and induce ovulation TZD improves TT, FT and SHBG improving the hormonal balance Eradication of leukemia stem cells in chronic myeloid leukemia patients by combining with tyrosine kinase inhibitors	**Advantages** Significant reduction in ALT, AST, fatty liver and lobular inflammation in patients with NASH in combination with Vitamin E Metformin combined with TZD was the best intervention to promote menstrual recovery: metformin combined with TZD could significantly increase ovulation rate Metformin combined with TZDs has been associated with better effects on increasing SHBG and promoting menstruation recovery than metformin alone Metformin combined with TZDs could significantly improve hirsutism and acne Metformin and pioglitazone appear to have beneficial anti-inflammatory effects in patients with MS and MetS **Limitations** Dominant negative PPARγ mutations lead to partial lipodystrophy: should be paid to regional fat increase and the development of lipomatous growths Increased edema in patients using pioglitazone due to effects on the renal tubular epithelial sodium channel and other effects in the collecting ducts, and possibly due to increased vascular permeability	Improvement in small dense lipoproteins with Vitamin E An increase in large LDL particles Favorable effects on TG and HDL-C Attenuation and stabilization of inflammation of atherosclerotic plaque and the alteration of the composition of the atherosclerotic nucleus Pioglitazone treatment, by lowering the level of PICP, may decrease the incidence of permanent atrial fibrillation in diabetic patients related to its inhibitory effect on the levels of AGEs

AGEs: Advanced Glycation End products; ALT: ALanine aminotransferase; AST: ASpartate aminotransferase; FT: Free Testosterone; HDL-C: High-Density Lipoprotein Cholesterol; LDL: Low-Density Lipoproteins; MetS: Metabolic Syndrome; MS: Multiple Sclerosis; NASH: Non-Alcoholic SteatoHepatitis; PICP: Peptide of type I Pro-Collagen; PPARγ: Peroxisome Proliferator-Activated Receptors gamma; RAS: Renin–Angiotensin System; SHBG: Sex Hormone Binding Globulin; SU: Sulfonylurea; T2DM: Type 2 Diabetes Mellitus; TG: Triglycerides; TT: Total Testosterone; TZDs: ThiaZoliDinediones.

In obese patients with polycystic ovary syndrome (PCOS), metabolic abnormalities related to IR and obesity are more significant than metabolic alterations caused by increased male sex hormones and anovulation [69]. TZDs such as pioglitazone and rosiglitazone are efficacious in improving IR and increased androgens in PCOS, whether as single therapy or combined with other antidiabetic agents [70]. Therapy with TZD improves peripheral insulin sensitivity and improves the effects of insulin on skeletal muscle and adipose tissue but not on insulin release [71]. Biguanide in association with TZD was the best intervention to promote menstrual recovery: metformin in association with TZD would greatly increase ovulation process [72] and pioglitazone could also recover menstrual cycles and cause ovulation [71]. Metformin in association with TZDs has been linked with improved effects on increasing Sex Hormone Binding Globulin (SHBG) and promoting menstruation recovery than metformin alone [70]. TZDs repress androgen biosynthesis in thecal cells, improving Total Testosterone (TT), Free Testosterone (FT) and SHBG better than metformin [73,74]. Moreover, metformin in association with TZDs could significantly ameliorate hirsutism and acne after 12 weeks of treatment [75].

Lipodystrophic syndromes constitute a diverse class of diseases represented by loss of subcutaneous adipose tissue, without undernourishment or increased catabolism [76]. The ability of adipose tissue to accumulate energy surplus increases, resulting in ectopic lipid accumulation (liver, muscle, kidney, pancreas) [76,77]. There is also chronic leptin reduction. Ectopic intramuscular and hepatic fat contributes to the progress of IR with early onset of diabetes mellitus, severe hypertriglyceridemia, NAFLD and a picture similar to that of PCOS [77]. Thiazolidinediones have been used in partial forms of LiPodystrophy (PL), but their use is debated [78]: they may ameliorate metabolic complications but should be cautiously prescribed in generalized lipodystrophy [79]. Lipodystrophy syndromes are rare and due to body fat deficiency, often associated with potentially serious metabolic complications, including diabetes, hypertriglyceridemia and steatohepatitis, or with viral infections (HIV) or injectable drugs. The guidelines for the management of this rare and heterogeneous pathology provide for a diagnosis of the clinical phenotype, supplemented, in some cases, by genetic tests; patients should be screened annually for diabetes, dyslipidemia, and liver, kidney and heart disease. Diet and metreleptin therapy are effective for metabolic complications in patients with generalized lipodystrophy and partial lipodystrophy, but other non-specific treatments for lipodystrophy, including metformin and glitazones for diabetes as well as statins, fibrates and proprotein convertase subtilisin/kexin type 9-inhibitors for dyslipidemia and coagulation abnormalities, may also be useful [80,81].

Thiazolidinediones may improve metabolic complications in partial lipodystrophy but should only be used with caution in generalized lipodystrophy according to guidelines. In patients with partial lipodystrophy, thiazolidinediones have improved HbA_1c_, triglycerides, liver volume, and steatosis, but may increase regional fat excess [82]. In NAFLD not associated with lipodystrophy, pioglitazone [59] showed the most consistent benefit for hepatic histopathology.

Several scientific studies had highlighted the increased risk of bladder cancer associated with the intake of pioglitazone: the increased risk exists, however, the conditions for avoiding the use of the drug are present only in a limited category of patients; it remains unclear whether the effect of the drug is premature or a consequence of prolonged use.

## 4. The Clinical Efficacy of Thiazolidinediones

Pioglitazone was introduced in the treatment of diabetes about 20 years ago and acts at the level of the nucleus by modulating the activation or suppression of many genes. Its durability in controlling blood sugar lasts for more years than sulfonylureas. Pioglitazone improves serum lipids, especially triglycerides and HDL-C. It has an anti-inflammatory and antithrombotic action, and has demonstrated important cardiovascular benefits in subjects with myocardial infarction or stroke, even in those without diabetes. In DM subjects with heart attack or stroke, pioglitazone is particularly indicated and is recommended by scientific societies. A prospective, randomized, double-blind trial assessed the effect of pioglitazone treatment in the evolution of unstable atrial fibrillation (AF) to stable atrial fibrillation in T2DM and to ascertain the mechanisms of any beneficial effect [83]. The association between diabetes and AF exists and is significantly greater than in the non-diabetic population as oxidative stress and inflammation were involved in the pathogenesis of AF. Among the various hypoglycemic therapies, there are studies that show a protective action performed by TZDs with anti-inflammatory and antioxidant effects as well as electrophysiological and structural atrial remodeling in addition to their antidiabetic activity. By lowering the terminal carboxy Peptide of type I Pro-Collagen (PICP), pioglitazone decreased the appearance of permanent AF in subjects with diabetes [83]. This is largely attributable to the fact that it reduced Advanced Glycation End products (AGEs) concentration [83].

The IRIS (Insulin Resistance Intervention after Stroke) study showed how pioglitazone can reduce the incidence of cardiovascular events (AMI or stroke) and diabetes (DM) in subjects with insulin resistance (IR) after a recent stroke or TIA; pioglitazone is shown to provide a cardiovascular benefit without increasing the risk of heart failure by monitoring side effects and modulating the dosage of therapy [84]. Pioglitazone suppresses the expression of transforming growth factor-β1 (TGF-β1) and tumor necrosis factor-α (TNF-α) in atrial tissue, molecules that are mediators of inflammation related to the incidence of AF mediated by fibrosis [85]. Pioglitazone effectively attenuates inflammatory profibrotic signals and vulnerability to AF possibly through its suppression in the expression of the monocyte chemotactic protein (MCP-1) [86]. Pioglitazone prevents age-related arrhythmogenic atrial remodeling and the incidence of AF by improving heat shock protein (HSP) 70 expression and antioxidant capacity and inhibiting the mitochondrial apoptotic signaling pathway [87].

Glitazones likely protect against Parkinson’s Disease (PD) in diabetic patient [88]. The mechanisms underlying the possible neuroprotective effect of GTZ are being evaluated. Probably glitazone acts by improving mitochondrial function, stimulating PPARγ, which, through the PPARγ 1-α coactivator pathway, leads to an increase in mitochondrial biogenesis [89]. It is possible that GTZ drugs ameliorate these defects by increasing mtDNA synthesis and overall mitochondrial mass. GTZ may also have anti-inflammatory action through reduced activation of microglia and inhibition of tumor necrosis factor α, as well as reduced nitric-oxide-mediated toxicity [90].

Glitazones could help eradicate leukemia stem cells in chronic myeloid leukemia subjects by combining with inhibitors of tyrosine kinase [91]. PPARγ is a nuclear receptor that acts as a transcription factor involved in the regulation of energy metabolism, cell cycle, cell differentiation and apoptosis. However, due to the pleiotropic effects of PPARγ activation in normal and tumor cells, PPARγ ligands interact with many antitumor treatment modalities and synergistically enhance their efficacy. Activation of the combination of PPARγ ligands with tyrosine kinase inhibitors (TKI) in chronic myeloid leukemia (CML) sensitizes leukemic stem cells to chemotherapy [92]. Moreover, this combination is believed to be the first drug therapy that can cure CML patients. Leukemia cells are highly dependent on high levels of the Signal Transducer and Activator of Transcription 5 (STAT5) protein. Tyrosine kinase inhibitors are unable to reduce STAT5 levels in quiescent stem cells, while pioglitazone activates the PPAR-gamma receptor and STAT5 is found downstream in the PPAR-gamma pathway. Activating the PPAR-gamma receptor with pioglitazone reduces STAT5 levels to the point where leukemic stem cells cannot survive [91].

Metformin and pioglitazone have been shown to possess a positive anti-inflammatory effect in subjects with Multiple Sclerosis (MS) and Metabolic Syndrome (MetS) [93]. This was associated with reduced leptin and higher adiponectin [93]. Improvements were also seen in the number and regulatory properties of T lymphocytes [93]. Patients on metformin or pioglitazone exhibited significantly fewer new lesions or enlargement of existing lesions on brain Magnetic Resonance Imaging (MRI) [91]. In addition, fasting blood glucose levels, insulin resistance, HbA_1c_, serum lipids and systolic blood pressure dropped significantly in both the metformin and the pioglitazone groups after 12 months [93].

The emergence of TZDs for diabetes prevention following GDM and in PCOS, the metabolic syndrome, and lipodystrophies is a new treatment perspective and could replace the use of metformin or other hypoglycemic agents in view of the beneficial effects seen in the clinical studies thus far. Pathologic states that could be treated with a hypoglycemic drug with pleiotropic effects, such as a glitazone, are numerous. There are potential pharmacological combinations of glitazones with GLP1-RAs/SGLT2is and/or nutraceuticals for the treatment of hepatic steatosis and neurodegenerative diseases.

## 5. Discussion

Diabetes treatment should aim to reach glycemia concentrations as close to normal as possible and the involvement of impaired alpha cell function has been suggested as playing an important role, since hyperglucagonemia is present in both Type 1 and T2DM [94]. When starting a treatment with the glitazones, it is useful to implement a broader, multidisciplinary approach to achieving glycemic goals with appropriate blood glucose monitoring. Intervention studies in subjects at risk of diabetes have shown that treatment with glitazones reduces progression from prediabetes to diabetes and increases the return to normal glucose tolerance; however, the prevention effect does not seem to persist after discontinuation of the treatment.

To date, however, neither the optimal therapeutic location of glitazones in T2DM, nor their long-term risks or their effects on diabetes complications are clear. In addition to their anti-hyperglycemic properties, glitazones exert effects on numerous cardiovascular risk factors (fibrinogen, C reactive protein, microalbuminuria, etc.); in the PROACTIVE study, pioglitazone therapy induced a significant diminution (−16%) of the secondary endpoint (all causes mortality, heart attack, stroke) but not of the primary one; thus, the TZDs effect on cardiovascular risk is not well-defined. In T2DM, glitazones are indicated as single-agent therapy (if intolerant to metformin) or in association with metformin and the sulphonylureas; they can also be used in triple therapy with them. The combination of pioglitazone and insulin requires close surveillance for fluid retention and the possible development of decompensated heart failure. In obese or overweight subjects, the glitazones–metformin combination should represent a preferential choice compared to that of the secretagogues (sulfonylureas, glinides) and metformin.

Data from clinical trials aimed at evaluating the efficacy of combination therapy with metformin/pioglitazone/exenatide versus sequential addition of metformin followed by glipizide and insulin in patients with long-standing type 2 diabetes mellitus (T2DM) have recently been recently published. A substantial decrease in HbA_1c_ and a three-fold increase in insulin sensitivity and a 30-fold increase in cell function were observed. Combination therapy metformin/pioglitazone/exenatide improving insulin sensitivity and β-cell function in patients with new-onset T2DM is a rather effective and safe therapeutic option that produces a greater and more lasting reduction in HbA_1c_. [95].

There are conflicting views on the issue of the cardiovascular safety of glitazones; as already written above, in a population at low cardiovascular risk, and in patients who have not had any events, the benefits of pioglitazone may be too mild to be found in absolute terms; the therapeutic strategies, metformin + sulphonylurea and metformin + pioglitazone, appear to be equivalent in terms of advantage over the primary goal of mortality and cardiovascular events, but treatment with pioglitazone has shown additional benefits in terms of the durability of glycemic control and the frequency of hypoglycemic cases. Unlike metformin which modulates blood glucose by reducing excessive glucose production in the liver and increasing glucose uptake by peripheral tissues, glitazones improve insulin sensitivity of peripheral tissues and the liver by regulating the expression of genes which at the cellular level control the metabolism of carbohydrates and lipids. There is also evidence that pioglitazone is able to reduce levels of pro-inflammatory cytokines, such as resistin [96], which has a significant role in cardiovascular diseases, diabetes and the metabolic syndrome [97].

The pivotal studies of glitazones had shown greater efficacy versus placebo and “non-inferiority” compared to other oral hypoglycemic agents, on the reduction of glycated hemoglobin levels and glycemia, without there being any evidence of a real ability to reduce vascular long-term complications. Fortunately, today the studies on the cardiovascular end-point have shown a possible “non-inferiority” compared to other drugs in secondary prevention. However, it has not yet been demonstrated whether metformin, the drug of first choice in the treatment of type 2 diabetes mellitus, has the same effects. Furthermore, as already mentioned, in patients with metabolic syndrome who are at high cardiovascular risk, the use of glitazones reduces comorbidities by reducing the cardiovascular risk.

Further, it should be pointed out that proper management of T2DM and cardiometabolic disorders is currently of greater importance during the COVID-19 pandemic, since they are at a higher risk of complications and mortality [98]. In addition, an increase in diabetic complications and cardiovascular deaths have been reported from the indirect effects of the pandemic, such as reduced access to healthcare facilities secondary to lockdowns and social isolation [99]. It is imperative to utilize all the available strategies, including optimizing medication use, to mitigate the effects of hyperglycemia on the health of our patients.

## 6. Conclusions

PPAR-gamma agonists exert effects on the liver, adipose tissue and the muscle and this has many consequences, such as reduced insulin resistance in the liver and peripheral tissues, decreased hepatic gluconeogenesis, and reduced blood glucose levels and HbA_1c_ (and other metabolic benefits). Evidence suggests that thiazolidinedione modification of the function of the mitochondrial target may contribute to lipid lowering and/or antidiabetic actions [100,101]. Although TZDs have not found as wide an acceptance for the therapy of type 2 diabetes as they deserve, their pleiotropic actions make them particularly appealing. Evidence has accumulated for their effectiveness beyond diabetes and into the arena of prediabetes, metabolic syndrome, PCOS, lipodystrophies, and for the prevention of diabetes itself. With their time-tested usefulness, these agents could see a resurgence in popularity for these and other therapeutic possibilities, particularly in combination therapy regimens. Their benefits should be carefully weighed against the risks, considering that the durability of glycemic beneficial actions appear to be sustainable. Rosiglitazone is out of the market in many countries due to an adverse cardiovascular risk profile [102] and pioglitazone seems to have no major contra-indications to its use with the exception of the potential risk of heart failure in susceptible individuals.

## Figures and Tables

**Figure 1 medicina-58-01475-f001:**
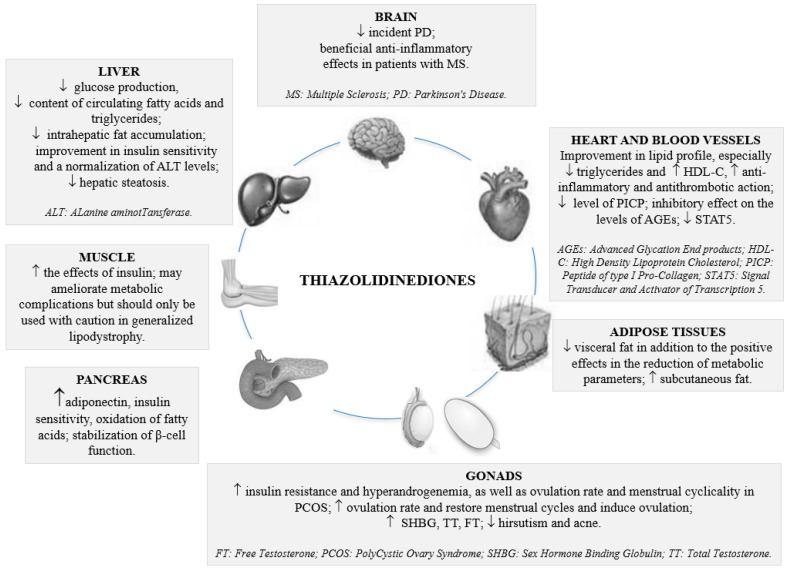
Pleiotropic effects of thiazolidinediones.

## Data Availability

Not applicable.

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
