# Peer review of "An Update on the Current and Emerging Use of Thiazolidinediones for Type 2 Diabetes"

_medicina, 2022, doi:10.3390/medicina58101475_

Round 1
Reviewer 1 Report
Dear All:
Receive greetings and congratulations for the submitted manuscript, comment on the relevant contribution you make to the knowledge of the pharmacological treatment of diabetes, I only suggest including the following reference among the citations you could make in the introduction:
The role of the α cell in the pathogenesis of diabetes: A world beyond the mirror. International Journal of Molecular Sciences. 10.3390/ijms22179504
María Sofía Martínez, Alexander Manzano, Luis Carlos Olivar, Manuel Nava, Juan Salazar, Luis D’marco, Rina Ortiz, Maricarmen Chacín, ,Clímaco Cano, Valmore Bermúdez, Angarita,L
Reviewer 2 Report
This review is about a potentially important drug and drug class which after its setback (related only to rosiglitazone) recently regained interest in the scientific community and perhaps will be revised once.
Therefore I found this paper basically inetesrting that could deserve its place and be published, however I found some points to be addressed.
These are as follows:
Authors discuss the imprortance of factors in IR on page 2, line 54-59. However the importance of NAFLD or with newer terminology MAFLD is missing, despite it is one of the most curcial factors and its role in T2DM development seems to be more preponderant than for example the kidneys. Therefore I would recommend to complete this list and add role of the liver (and the significance of fatty liver), with citing at least one review on this topic (e.g.: [PMID: 25083080 DOI: 10.3748/wjg.v20.i27.9072]
In table 1 the full trial name of the Pivens trial is wrongly indicated, please correct (p3)
The discussion is also slightly vague in the sense that it starts with:
„PPAR-gamma agonists, represented predominantly by the TZDs, are medications that act by ameliorating insulin resistance in muscle and other tissues. „
Why did the authors outline only the muscle tissue?
It would be more appropriate to state that PPARg agonists have effect on the liver, adipose tissue and the muscle and this has many consequences , such as the reduction of insulin resistance in the liver and peripheral tissues, decrease of hepatic gluconeogenesis, and reduction of blood glucose levels and HbA1c (and other metabolic benefits)
In addition the bidning of pioglitazone to the outer mitochondrial membrane ( doi:10.1152/ajpendo.00424.2003. PMID 14570702 , . doi:10.1073/pnas.0707189104.PMID 17766440) is missing from the text.
I would also recommend a moderate language polishing:
Please consider the use „factors” instead of „actors” (page 2, line 53) due to that it is a more scientific term, etc.
